# Canine Somatic Mutations from Whole-Exome Sequencing of B-Cell Lymphomas in Six Canine Breeds—A Preliminary Study

**DOI:** 10.3390/ani13182846

**Published:** 2023-09-07

**Authors:** Sungryong Kim, Namphil Kim, Hyo-Min Kang, Hye-Jin Jang, Amos Chungwon Lee, Ki-Jeong Na

**Affiliations:** 1Laboratory of Veterinary Laboratory Medicine, College of Veterinary Medicine, Chungbuk National University, Cheongju 28644, Republic of Korea; vet08dannyk@gmail.com (S.K.); hm.chobi@gmail.com (H.-M.K.); 2Biophotonics and Nano Engineering Laboratory, Department of Electrical and Computer Engineering, Seoul National University, Seoul 08826, Republic of Korea; namphilkim97@gmail.com; 3Department of Biomedical Laboratory Science, Daegu Health College, Daegu 41453, Republic of Korea; hjjang@dhc.ac.kr; 4Meteor Biotech, Co., Ltd., Seoul 08826, Republic of Korea; amos.lee@meteorbiotech.com

**Keywords:** canine lymphoma, whole-exome sequencing, B-cell, PARR

## Abstract

**Simple Summary:**

The aim of this study was to identify somatic mutations in dogs with B-cell lymphoma (BCL) using whole-exome sequencing (WES) and to investigate the impact of variants from lymph node (LN) aspirate samples compared with whole blood (WB) samples. This study analyzed DNA samples from eight dogs with BCL and conducted immunophenotyping using PCR for antigen receptor rearrangement (PARR). DNA was extracted and sequenced, and variant calling was performed. The analysis revealed highly common somatic variants, including a variant in the Golgi integral membrane protein 4 (GOLIM4) gene, which is associated with the endosome-to-Golgi protein trafficking pathway. Other notable variants were identified in genes such as desmocollin1 (DSC1), lipoxygenase homology domains 1 (LOXHD1), and glycoprotein VI platelet (GP6). The results suggest potential genetic markers and pathways involved in BCL in dogs. This study provides valuable insights into the genomic landscape of BCL in dogs, contributing to our understanding of the disease and potentially facilitating the development of targeted therapies in veterinary medicine.

**Abstract:**

Canine lymphoma (CL) is one of the most common malignant tumors in dogs. The cause of CL remains unclear. Genetic mutations that have been suggested as possible causes of CL are not fully understood. Whole-exome sequencing (WES) is a time- and cost-effective method for detecting genetic variants targeting only the protein-coding regions (exons) that are part of the entire genome region. A total of eight patients with B-cell lymphomas were recruited, and WES analysis was performed on whole blood and lymph node aspirate samples from each patient. A total of 17 somatic variants (*GOLIM4*, *ITM2B*, *STN1*, *UNC79*, *PLEKHG4*, *BRF1*, *ENSCAFG00845007156*, *SEMA6B*, *DSC1*, *TNFAIP1*, *MYLK3*, *WAPL*, *ADORA2B*, *LOXHD1*, *GP6*, *AZIN1*, and *NCSTN*) with moderate to high impact were identified by WES analysis. Through a Kyoto Encyclopedia of Genes and Genomes (KEGG) pathway analysis of 17 genes with somatic mutations, a total of 16 pathways were identified. Overall, the somatic mutations identified in this study suggest novel candidate mutations for CL, and further studies are needed to confirm the role of these mutations.

## 1. Introduction

Advances in next-generation sequencing technologies have made it easy and inexpensive to generate large amounts of genomic data. Determining the sequence of the entire genome is called whole-genome sequencing (WGS), and determining the sequence of the entire exon is called whole-exome sequencing (WES) [1]. Almost all protein-coding genes have discontinuous structures. Protein-coding regions are fragmented into several species called exons [2]. Exomes represent only approximately 1% of the genome; therefore, WES is less expensive than WGS [3]. WES technology is a proven method for identifying functionally relevant genetic variants in diseases such as cancers [4].

Studies using WES in human lymphomas have shown that significantly mutated genes such as *CD79B*, *TP53*, *CARD11*, *MYD88*, and *EZH2* are associated with large B-cell lymphoma (BCL) [5,6,7]. A study using WES in three breeds of dogs (Cocker Spaniel, Golden Retriever, and Boxer) predisposed to canine lymphoma (CL) confirmed mutations in *TRAF3-MAP3K14*, *FBXW7*, and *POT1* [8]. The mutations identified using WGS studies in CL were in *ST6GALNAC5*, *ENSCAFG00000007370*, *PPP2CB*, *TP53*, *SH2B3*, *ZNF503*, *SETD2*, and *COX18* [9].

The purpose of this study is to identify somatic mutations in dogs with BCL by pairwise WES of DNA from eight dogs to determine the impact of variants from lymph node (LN) aspirate samples compared with whole blood (WB) samples. In addition, a bioinformatic Kyoto Encyclopedia of Genes and Genomes (KEGG) pathway analysis using somatic variant genes identified in this study was performed to elucidate their roles in CL.

## 2. Materials and Methods

### 2.1. Samples

From 2016 to 2020, among the samples requested for cytological and PCR for antigen receptor rearrangements (PARR) tests for the diagnosis of lymphoma at the Laboratory of Veterinary Laboratory Medicine, Chungbuk National University, samples containing WB were used for WES analysis. The inclusion criteria for the study were multicentric lymphoma with enlarged LNs and no previous or current cancer diagnosis other than lymphoma. Of the eight subjects enrolled in this study, two were Maltese, two were Welsh Corgis, one was a Cocker Spaniel, one was a Shih Tzu, one was a White Terrier, and one was a mixed breed. The detailed information of the eight dog patients is presented in Table 1. All had BCL as determined by cytology and PARR.

### 2.2. Immunophenotyping

Immunophenotyping of all samples was determined by PARR. DNA was extracted from LN aspirates using a MagPurix^®^ Tissue DNA Extraction Kit and MagPurix^®^ 12s automated nucleic acid purification system (Zinexts Life Science Corp., New Taipei City, Taiwan), according to the manufacturer’s instructions. DNA preparations were stored at −80 °C until use. For this extraction, fine-needle aspiration (FNA) needles were washed with 600 μL of phosphate-buffered saline. The collected samples were vortexed shortly, and 200 μL was used for extraction. The final volume of the DNA elution was 50 μL. PCR was performed using primers used for amplification of Cµ (positive control), TCRγ CDR3, and Ig CDR3 (Table 2), as previously described [10]. Using distilled water, a negative control was run on each sample to ensure that no contamination was present. The amplification was performed in TaKaRa PCR Thermal Cycler Dice^®^ Touch (TaKaRa Bio, Shiga, Japan). The PCR reaction conditions for each product (Cµ, IgH major, IgH minor, and TCRγ) are shown in Table 3. The PCR products were separated by electrophoresis and detected using the Qsep 100 automatic nucleic acid protein analysis system (BiOptic, New Taipei City, Taiwan). The DNA samples were processed with a premade cartridge supplied with a DNA size marker and dye (Standard Cartridge Cat. No: C105201). According to the manual, all peaks and alleles were analyzed using Qsep 100 software.

### 2.3. DNA Extraction and Sequencing

DNA was extracted from WB and LN aspirates using the MagPurix Blood DNA Extraction kit and MagPurix Tissue DNA extraction kit (Zinexts Life Science Corp., New Taipei City, Taiwan), respectively, according to the manufacturer’s instructions. The sample quantity and purity were assessed with a NanoDrop spectrophotometer (260/280: 1.6–2.3, 260/230 > 1.6), electrophoresis (No DNA degradation), and Qubit fluorometric quantitation (≥100 ng/μL). Extracted DNA samples were sent to Theragen Bio (Seongnam, Republic of Korea) for NovaSeq 6000 (Illumina Inc., San Diego, CA, USA) sequencing with qPCR and SureSelect XT Canine All Exon V2 kit (Agilent, Santa Clara, CA, USA).

### 2.4. Variant Calling

The raw sequencing data in the FASTQ format were subjected to a quality check stage using FastQC v0.10.1 [11] program, and the adapter sequences were removed using Cutadapt v1.81 [12]. The processed data were aligned to a canine reference genome from the Boxer breed (CanFam3.1) using BWA v0.7.17 [13] and were sorted and marked for duplicates using Picard. Somatic variants were called using VarScan v2.3.9 using the alignment data from blood or oral samples as the control. Variant effects were predicted using VEP Ensembl web interface v107 [14] with ROS Cfam 1.0 as the reference. Variant effects such as LOF, disruptive insertions and deletions, stop gain and loss, and splice region variants were classified as high-impact variants, while missense variants, untranslated region (UTR) variants, in-frame insertions, and deletions were classified as moderate-impact variants. Synonymous, intron, and intergenic variants were classified as low-impact variants [9]. Variants with less than or equal to 5% variant frequency in blood samples and greater than or equal to 15% variant frequency in tumor samples were considered tumor-specific according to the VarScan guidelines [15], while variants with greater than or equal to 15% variant frequency in blood were classified as germline variants. A schematic diagram of the pipeline analysis is shown in Figure 1.

### 2.5. Sharing Analysis

Identification of individual variants and genes targeted by said variants that were commonly seen among subjects was performed using an in-house Python 3.9 code that uses VEP files as input.

### 2.6. Protein–Protein Interaction (PPI) Network Construction and Analysis of Modules

The PPI network of differentially expressed genes (DEGs) was obtained from the Search Tool for the Retrieval of Interacting Genes/Proteins (STRING, Version 10.0; https://www.strind-db.org/, accessed on 21 June 2023) online database, and a confidence score ≥ 0.4 with FDR stringency of 5% was set as the cutoff criterion, which is the default setting. The PPI network was imported into Cytoscape 3.10.0 (https://cytoscape.org/, accessed on 21 June 2023) software for network visualization.

### 2.7. KEGG Pathway Analysis

The gene names that were obtained in this study were converted to KEGG gene IDs, according to UniProt (https://uniprot.org/, accessed on 19 December 2022). Then, the KEGG pathway analysis was performed on a KEGG mapper (https://www.genome.jp/kegg/mapper/color.html, accessed on 20 December 2022) using the obtained KEGG gene IDs.

## 3. Results

### 3.1. Variant Level Analysis Reveals Highly Shared GOLIM4 Variant

Out of the myriad of somatic variants identified within each subject, variants with a higher likelihood of being directly associated with BCL were selected because the chances of a random mutation appearing at the exact location in the exact same way in multiple subjects are slim. Among 480 variants that were shared by at least two subjects, 19 were classified as moderate- or high-impact variants by VEP. By mapping these shared somatic variants onto the chromosome structure of canines, we identified multiple chromosomes with multiple somatic mutations shared by more than three subjects. In particular, chromosome 25 had consecutive highly shared variants near its edge, and chromosome 34 had a somatic variant shared by half of the subjects (Figure 2).

We also attempted to identify shared somatic variants with moderate to high predicted impact and the genes that they target. We found that the highly shared variant in chromosome 34 was a variant in the 3′ UTR region of the gene Golgi integral membrane protein 4 (*GOLIM4*), which plays a role in the endosome-to-Golgi protein trafficking pathway. Other notable variants include frameshift variants in gene desmocollin1 (*DSC1*), lipoxygenase homology domains 1 (*LOXHD1*), and glycoprotein VI platelet (*GP6*); a start lost variant in gene Adenosine A2B receptor (*ADORA2B*); and a stop lost variant in the gene antizyme inhibitor 1 (*AZIN1*) (Table 4).

### 3.2. Gene Level Sharing Analysis Shows Variant Accumulation in a Specific PPI Network

After investigating the individual shared variants, we investigated genes that were commonly targeted by somatic variants in multiple subjects, although they were not identical variants. We identified 2131 genes that were commonly targeted, and more than 150 genes were shared by more than half of the subjects. Some examples of chromosomes containing many of the highly shared genes include chromosomes 1, 2, 13, and 15 (Figure 3a). By filtering out 48 genes that were commonly targeted by somatic variants with high or moderate impact, we observed that *GOLIM4* was the target of many more 3′ UTR variants that differed from the previously identified locus. Other genes that were highly shared include wings apart-like protein homolog (*WAPL*), which also had 3′ UTR variants, and C-C motif chemokine ligand 23 (CCL23), which had missense and 3′ UTR variants. (Figure 3b).

To identify PPI networks regarding the shared variants, we analyzed 333 highly shared annotated genes targeted by tumor variants that were shared by greater than or equal to half of the subjects and found that there was a large network of interconnected proteins with over 50 nodes that were affected by them as well as additional smaller networks (Figure 3c).

### 3.3. KEGG Pathway Analysis Results

As a result of KEGG pathway analysis of 17 genes with somatic mutations, a total of 16 pathways (vascular smooth muscle contraction, calcium signaling pathway, platelet activation, ECM-receptor interaction, oxytocin signaling pathway, regulation of actin cytoskeleton, axon guidance, apelin signaling pathway, alcoholism, cGMP-PKG signaling pathway, gastric acid secretion, Alzheimer’s disease, focal adhesion, notch signaling pathway, neuroactive ligand-receptor interaction, and Rap1 signaling pathway) were identified (Table 5). After excluding pathways associated with only one gene, the following pathways were identified: vascular smooth muscle contraction, calcium signaling pathway, and platelet activation (Figure 4, Figure 5 and Figure 6).

### 3.4. Highly Shared Germline Mutations in Diffuse Large BCL (DLBCL)-Associated Genes and Pathways

Analysis of germline variants present in 12 known DLBCL-associated genes (*MYC*, *BCL2*, *BCL6*, *EZH2*, *GNA13*, *PTEN*, *TP53*, *TNFAIP3*, *PRDM1*, *CD79B*, *CARD11*, and *MYD88*) [5,6,7,16,17,18,19] and their PPI networks annotated via string DB revealed several stereotypic moderate- to high-impact variants (Figure 7). Among these cases, TRRAP, EP300, USP7, and ATM showed variants that could greatly impact the function of proteins, such as stop gains, stop losses, and frameshifts. In particular, for USP7, the same germline variant causing a stop gain was present in five of the eight subjects. Several missense variants, including those present in TAX1BP1, PRMT5, and SYK, were shared by all or seven out of the eight subjects involved, indicating a strong possibility that such mutations in the germline may be associated with DLBCL.

## 4. Discussion

In this study, WES analysis was conducted on both WB and LN aspirate samples from eight dogs with BCL. Based on the results, a comparison between LN aspirate and WB revealed the presence of mutations in LN aspirate samples. This led to the identification of 17 shared somatic variants with moderate or high impact, including *GOLIM4*, *ITM2B*, *STN1*, *UNC79*, *PLEKHG4*, *BRF1*, *ENSCAFG00845007156*, *SEMA6B*, *DSC1*, *TNFAIP1*, *MYLK3*, *WAPL*, *ADORA2B*, *LOXHD1*, *GP6*, *AZIN1*, and *NCSTN*. According to the literature, considering the functions of these genes and related diseases, *GOLIM4*, *ITM2B*, *STN1*, *DSC1*, *TNFAIP1*, *WAPL*, and *NCSTN* can be assumed to play roles as tumor suppressor genes. GOLIM4 (also known as GPP130) is a component of the Golgi transport complex that plays an important role in the transport of Golgi proteins [20]. Although the Golgi apparatus may be involved in tumor biological processes, the function of GOLIM4 during tumorigenesis remains unclear. Nevertheless, the Golgi apparatus and endosome dysfunction are involved in the progression of various tumors, and increased expression of GOLIM4 has been shown to inhibit cancer cell proliferation, promote apoptosis, and induce G1 phase arrest in human head and neck cancer cell lines [21]. Integral membrane protein 2B (ITM2B; also known as BRI2) is a type II transmembrane protein that is a substrate for regulated intramembrane proteolysis [22]. ITM2B induces apoptosis and inhibits proliferation [23]. Downregulation of ITM2B has been identified in human lung cancer tissues; therefore, ITM2B appears to play a role as a tumor suppressor gene [24]. STN1, along with CTC1 and TEN1, is a component of the CST (CTC1-STN1-TEN1) complex and is responsible for maintaining telomere and genome integrity [25]. CST was first identified as a telomere-binding protein complex and functions in telomere replication and protection. CST can mediate end protection at double-strand breaks, likely using a similar strategy to fill in the telomeric C-strand. Supporting this observation, CST has been shown to promote polymerase inhibitor sensitivity in BRCA1-deficient cancer cells. Given its essential roles in replication and DNA repair, CST is known to be important for genome stability [26]. DSC1 was predicted to encode a sodium channel based on a high sequence similarity with vertebrate and invertebrate sodium channel genes. In human medicine, decreased expression of DSC1 was related to the poor differentiation and prognosis of head and neck squamous cell carcinoma, lung cancer, melanoma, and colorectal carcinoma [27,28,29,30]. TNF-α-induced protein 1 (TNFAIP1; B12) gene is a highly conserved gene in several species and is known as a tumor suppressor gene. *TNFAIP1* is induced by TNF-α and interleukin-6, and it is mainly involved in DNA synthesis and repair and apoptosis [31,32,33]. In human medicine, breast cancer, gastric carcinomas, and lung cancer are known to be associated with *TNFAIP1* mutations [34,35,36]. WAPL is important in regulating the level of aggregation of chromosomes by separating cohesive loops from chromatin. By separating cohesion from chromatin, WAPL is a regulator of the loading and unloading cycle. The loss of WAPL is known to result in p53-dependent cell cycle arrest [37,38]. The role of nicastrin (NCSTN) remains unknown. However, NCSTN is known to be related to AKT and p-AKT, which affect cell proliferation, growth, and differentiation [39]. In addition, incomplete expression of NCSTN is known to reduce the expression of miR-100-5p, which acts as a tumor suppressor involved in cell self-renewal and wound healing [40].

On the other hand, *BRF1*, *SEMA6B*, *ADORA2B*, *GP6*, and *AZIN1* can be assumed to function as oncogenes. The *BRF1* gene encodes the BRF1 protein in its zinc ribbon domain and directly participates in the process of protein synthesis. Deregulation of BRF1 is associated with cell proliferation, cell transformation, and tumorigenesis. BRF1 is overexpressed in hepatocellular carcinoma, breast cancer, gastric cancer, prostate cancer, and lung cancer in humans [41,42,43,44]. Semaphorin 6b (SEMA6B) is a member of the semaphoring axon-guidance family and was initially characterized as an axon guidance factor with axon navigation functions but has also been demonstrated to induce or inhibit tumor progression. The overexpression of this gene is related to colorectal cancer in humans [45,46,47]. *ADORA2B* encodes a protein belonging to the G protein-coupled receptor superfamily that plays a role in tissue distribution along with A1, A2A, and A3. Abnormal expression of ADORA2B may play a pathophysiological role in some human cancers [48]. ADORA2B is highly expressed in oral cancer, lung adenocarcinoma, and prostate cancer and promotes the proliferation and metastasis of carcinoma cells [48,49,50]. GP6 is a transmembrane protein that is the major signaling receptor for collagen on platelets and regulates several platelet functions, such as adhesion, aggregation, and procoagulant activity [51,52]. In addition, GP6 plays a role in supporting platelet adhesion to tumor cells, which is known to be involved in the metastasis of colorectal cancer and breast cancer [53]. Adenosine-to-inosine (A-to-I) RNA editing catalyzed by adenosine deaminases acting on RNA enzymes is a post-transcriptional modification that has emerged as a key player in tumorigenesis and cancer progression. AZIN1 has been identified as one of the most frequently occurring A-to-I RNA alterations in colorectal cancer and hepatocellular carcinoma and acts as an oncogene [54,55].

The function of three genes (*UNC79*, *PLEKHG4B*, and *ENSCAFG00845007156*) and their associations with cancers have not been clearly identified. UNC79 protein forms an NALCN complex with NALCN, FAM155, and UNC80 proteins, which are involved in voltage-gated sodium and calcium channels [56,57]. There is no research on the association between *UNC79* mutations and cancers. Mutations in the pleckstrin homology domain-containing family g member 4b (*PLEKHG4B*; puratrophin-1) gene are associated with the hereditary neurological disorder autosomal dominant spinocerebellar ataxia. However, the biochemical function of this gene product has not been described [58]. Moreover, there is no research about the association between *PLEKHG4B* mutations and cancers. *ENSCAFG00845007156* is similar to human aldo-keto reductase family 1 member D1, but its function has not been identified, and protein analysis has not been performed to date.

For both genes (*MYLK3* and *LOXHD1*), the function of each gene was not associated with cancers but was associated with diseases other than cancers. Myosin light chain kinase 3 (*MYLK3*) is a protein-coding gene that acts as a regulator of the actin cytoskeleton and immune response signaling. WES revealed that MYLK3 mutations are associated with dilated cardiomyopathy in humans [59]. *LOXHD1* encodes a protein consisting of 15 polycystin lipoxygenase α-toxin repeats, which can bind lipids and proteins in other proteins [60]. Mutations in *LOXHD1* cause progressive hearing loss [61]. As a result, mutations in tumor suppressor genes can affect cancers such as CL. Therefore, *GOLIM4*, *ITM2B*, *STN1*, *DSC1*, *TNFAIP1*, *WAPL*, and *NCSTN* mutations may be prognostic markers for patients with CL. However, the functions of *UNC79*, *PLEKHG4*, and *ENSCAFG00845007156* and their associations with cancers have not yet been identified. In addition, *MYLK3* and *LOXHD1* mutations appear to have a very low association with CL.

As a result of constructing a PPI network for approximately 330 genes targeting 480 shared somatic variants, one large network was formed. A total of seven genes (*STN1*, *AZIN1*, *ITM2B*, *ADORA2B*, *SEMA6B*, *NCSTN*, and *DSC1*) in the network were genes identified with moderate- to high-impact mutations. However, none of the many genes observed in the network appear to be directly related to CL.

KEGG pathway analysis revealed that three pathways are associated with at least two mutated genes. Among them, two pathways suspected to be closely related to BCL are the calcium signaling pathway and the platelet activation pathway. Processes such as cell proliferation and death and gene transcription are essential for regulating cellular functions, and tight regulation of calcium signaling is fundamental in this process [62]. Therefore, changes in calcium signaling can cause various diseases, including tumors, and these changes have been confirmed in cancer cell lines [63]. Platelet function plays an important role not only in hemostasis but also in tumor metastasis [64]. Some research has demonstrated reduced tumor metastasis after experimentally inducing thrombocytopenia in mouse models [65]. Therefore, mutations in genes (*ADORA2B*, *MYLK3*, and *GP6*) involved in both pathways may be mutations of interest in BCL.

Overall, considering the function of each gene, *GOLIM4*, *ITM2B*, *STN1*, *DSC1*, *TNFAIP1*, *WAPL*, and *NCSTN* mutations were found to be highly associated with BCL, and *ADORA2B*, *MYLK3*, and *GP6* mutations were suspected to be associated with BCL through KEGG pathway analysis. Therefore, *GOLIM4*, *ITM2B*, *STN1*, *DSC1*, *TNFAIP1*, *WAPL*, *NCSTN*, *ADORA2B*, *MYLK3*, and *GP6* mutations are proposed as candidate mutations associated with BCL.

There are some limitations in this study. First, the patients with BCL in this study were various canine breeds. Different breeds of dogs exhibit different physical characteristics, and these characteristics may be related at a genetic level. Second, the number of patients used in this study was small. The inclusion of more patients will more clearly identify commonly observed somatic variants. Third, a number of germline mutations were observed, but these could not be analyzed due to the vast amount of data. A review of germline mutations In human DLBCL suggests a possible association between germline mutations and lymphoma [66]; therefore, further studies on germline mutations and CL in more cases of neoplasm in dogs of the same breed will be needed.

## 5. Conclusions

This study demonstrates the utility of WES in identifying somatic mutations in dogs with BCL. The identification of shared variants and their associated genes contributes to the understanding of the molecular mechanisms underlying BCL development and progression in dogs. Further analysis using bioinformatic tools, such as PPI network construction and KEGG pathway analysis, may provide additional insights into the functional roles of these genes and their involvement in CL. These findings pave the way for future research focusing on targeted therapies and personalized medicine for dogs with BCL.

## Figures and Tables

**Figure 1 animals-13-02846-f001:**
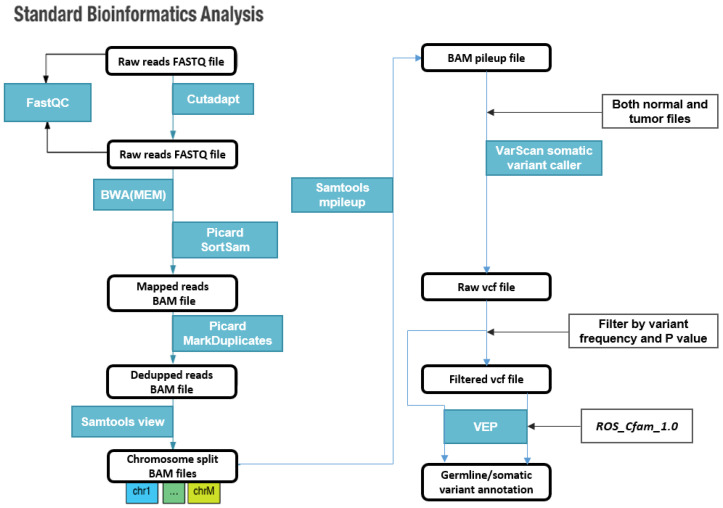
Standard bioinformatic analysis pipeline used in this study. The blue boxes show the name of the tools used.

**Figure 2 animals-13-02846-f002:**
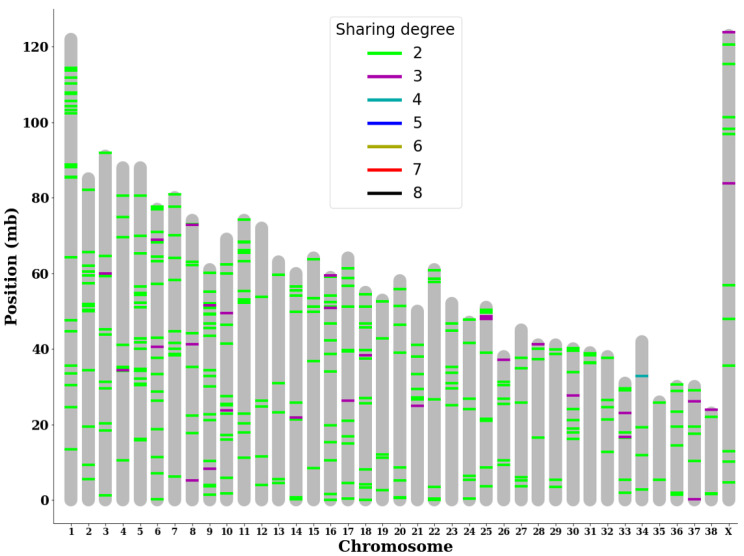
Analysis of somatic tumor variants shows specific highly shared loci. Individual somatic variants shared between at least two subjects were plotted based on the degree of sharing (number of subjects sharing said variant) and the location of each variant on the genome.

**Figure 3 animals-13-02846-f003:**
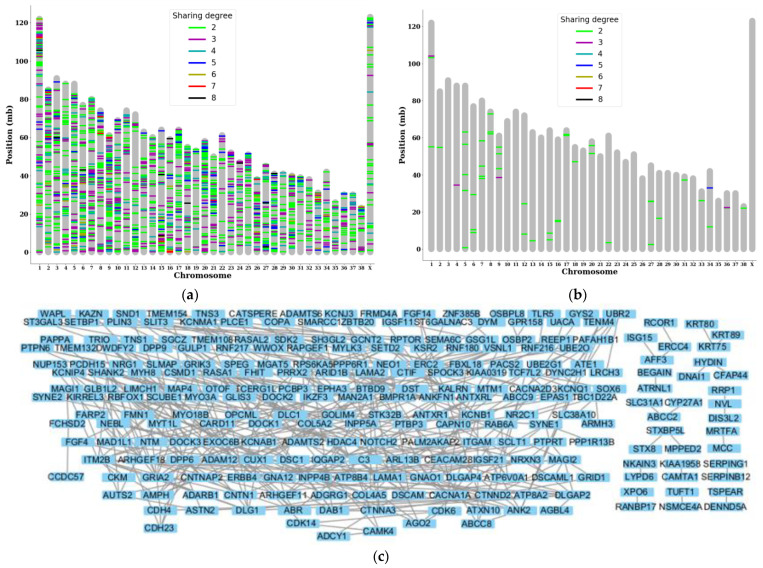
Gene-level analysis showing a major PPI network being subjected to somatic mutations: (**a**) Genes targeted by somatic mutations in multiple subjects were plotted based on the locus of the chromosome and the degree of sharing. (**b**) Genes targeted by somatic mutations with high or moderate impact in multiple subjects. (**c**) PPI network of the named genes targeted by shared somatic variants. Nodes with no connections were removed for enhanced visibility, and the built-in hierarchical layout of Cytoscape was used.

**Figure 4 animals-13-02846-f004:**
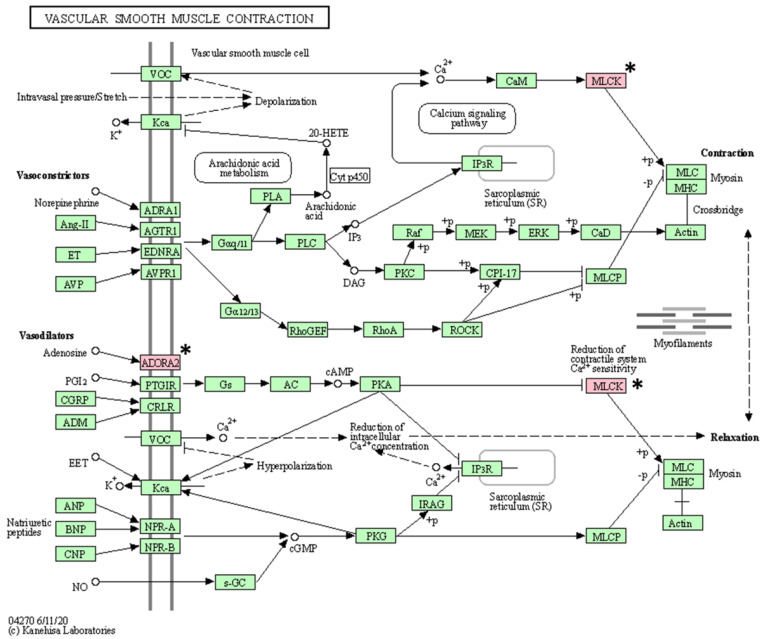
Vascular smooth muscle contraction pathway confirmed in the KEGG pathway analysis. The asterisk (*) indicates the genes associated with this pathway.

**Figure 5 animals-13-02846-f005:**
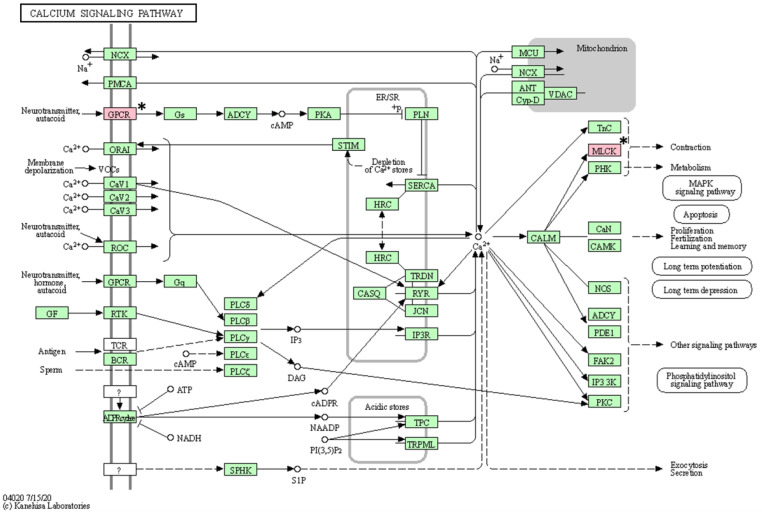
Calcium signaling pathway confirmed in the KEGG pathway analysis. The asterisk (*) indicates the genes associated with this pathway.

**Figure 6 animals-13-02846-f006:**
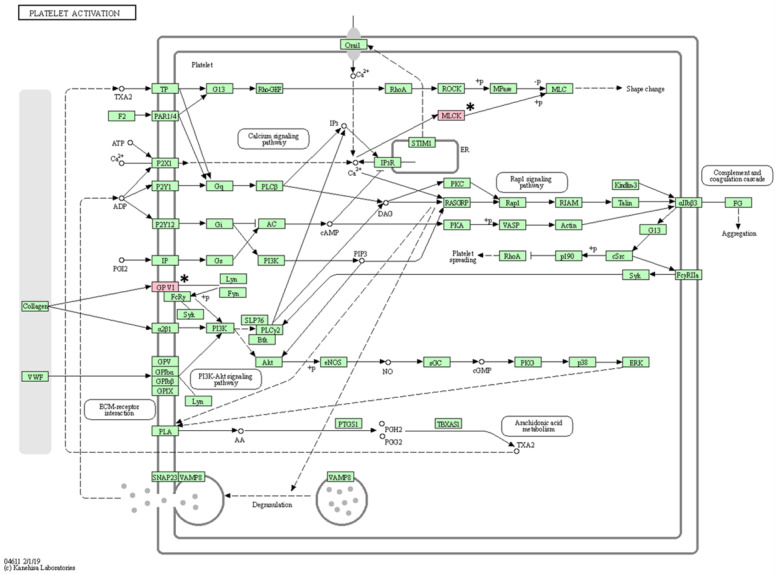
Platelet activation pathway confirmed in the KEGG pathway analysis. The asterisk (*) indicates the genes associated with this pathway.

**Figure 7 animals-13-02846-f007:**
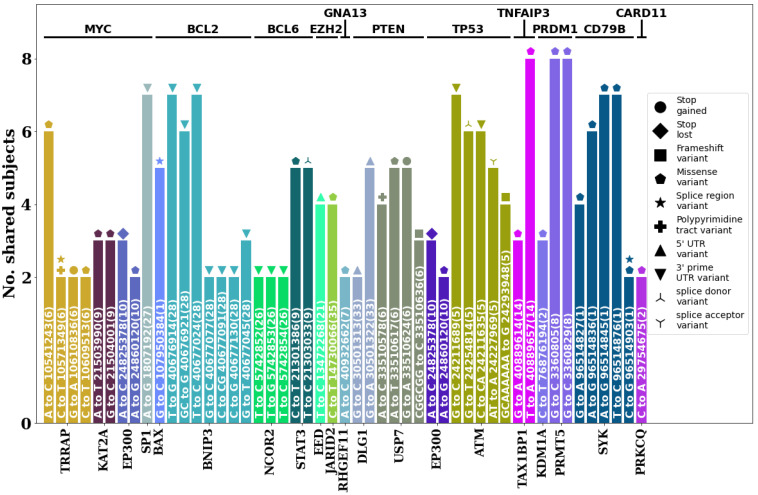
Stereotypic moderate- to high-impact germline mutations found in lymphoma subjects. The mutation, position, and chromosome based on CanFam3.1 are written within the bar plots with the VEP predicted effect shown as icons. Those with two or more icons have multiple effects assigned.

**Table 1 animals-13-02846-t001:** Clinical characteristics of dogs used in this study.

Sample No.	Breed	Age (Years)	Sex	Cytological Results	PARR Results (Monoclonal)	Lymphoma WHO Stage	Survival Time (Days)
1	Maltese	5	SF	Diffuse, large	IgH major	stage IV, substage a	Euthanasia
2	Welsh Corgi	8	IF	Diffuse, large	IgH major	stage IV, substage b	30, loss
3	Cocker Spaniel	11	IF	Diffuse, intermediate	IgH major	stage IV, substage b	Loss
4	Shih Tzu	10	CM	Diffuse, intermediate	IgH major	stage V, substage b	406
5	Maltese	6	SF	Diffuse, intermediate	IgH minor	stage IV, substage b	407
6	White Terrier	6	IF	Diffuse, large	IgH minor	stage IV, substage a	435, loss
7	Mixed breed	7	IM	Diffuse, large	IgH major	stage IV, substage a	-
8	Welsh Corgi	9	CM	Diffuse, intermediate	IgH major	stage IV, substage a	232

PARR, PCR for antigen receptor rearrangement; WHO, World Health Organization; SF, spayed female; IF, intact female; CM, castrated male; IM, intact male.

**Table 2 animals-13-02846-t002:** Primers used for amplification of Cµ (positive control), TCRγ CDR3, and Ig CDR3.

Reaction No.	Product	Primer Names	Primer Specificity	Primer Sequence
1	Cµ	Sigmf1	Cµ	TTC CCC CTC ATC ACC TGT GA
		Srµ3	Cµ	GGT TGT TGA TTG CAC TGA GG
2	IgH major	CB1	VH	CAG CCT GAG AGC CGA GGA CAC
		CB2	JH	TGA GGA GAC GGT GAC CAG GGT
3	IgH minor	CB1	VJ	CAG CCT GAG AGC CGA GGA CAC
		CB3	JH	TGA GGA CAC AAA GAG TGA GG
4	TCRγ	TCRγ1	JH	ACC CTG AGA ATT GTG CCA GG
		TCRγ2	JH	GTT ACT ATA AAC CTG GTA AC
		TCRγ3	VH	TCT GGG RTG TAY TAC TGT GCT GTC TGG

**Table 3 animals-13-02846-t003:** PCR reaction conditions used for this study.

Reaction No.	Product	Initial Denaturation	40 Cycles			Final Extension
			Denaturation	Annealing	Extension	
1	Cµ	94 °C, 15 s	94 °C, 15 s	57 °C, 15 s	72 °C, 15 s	72 °C, 15 s
2	IgH major	94 °C, 15 min	94 °C, 15 s	63 °C, 15 s	72 °C, 15 s	-
3	IgH minor	94 °C, 15 min	94 °C, 15 s	57 °C, 15 s	72 °C, 15 s	72 °C, 1 min
4	TCRγ	94 °C, 15 min	94 °C, 15 s	52 °C, 15 s	72 °C, 15 s	-

**Table 4 animals-13-02846-t004:** Moderate- or high-impact gene mutations found in patients with CL with existing, defined variant names.

Gene Name	Chromosome	Variant Type (Position Reference Base Variant Impact)	Degree of Sharing
*GOLIM4*	34	32880086_C_T_3_prime_UTR_variant	4
*ITM2B*	22	3396316_CTGGGGGCGGGTGGG_C_5_prime_UTR_variant	2
*STN1*	28	16517367_CA_C_splice_polypyrimidine_tract_variant&intron_variant	2
*PLEKHG4B*	34	11903730_G_C_missense_variant&splice_region_variant	2
*PLEKHG4B*	34	11903747_G_T_splice_polypyrimidine_tract_variant&intron_variant	2
*UNC79*	8	63061314_A_G_splice_polypyrimidine_tract_variant&intron_variant	2
*UNC79*	8	63061318_G_A_splice_polypyrimidine_tract_variant&intron_variant	2
*BRF1*	8	72881347_G_A_missense_variant	2
*ENSCAFG 00845007156*	16	15352766_C_G_missense_variant	2
*SEMA6B*	20	55831203_G_A_3_prime_UTR_variant	2
*DSC1*	7	58326567_CT_C_frameshift_variant	2
*TNFAIP1*	9	43411403_T_G_missense_variant	2
*MYLK3*	15	8510405_G_A_splice_donor_5th_base_variant&intron_variant	2
*WAPL*	4	34552139_TTC_T_3_prime_UTR_variant	2
*ADORA2B*	5	40018212_CA_C_frameshift_variant&start_lost	2
*LOXHD1*	7	44757742_GAA_G_frameshift_variant	2
*GP6*	1	103278325_A_AG_frameshift_variant	2
*AZIN1*	13	4527111_A_G_stop_lost	2
*NCSTN*	38	21997405_T_G_missense_variant	2

*GOLIM4*, Golgi integral membrane protein 4; *ITM2B*, integral membrane protein 2B; *PLEKHG4B*, pleckstrin homology domain-containing family g member 4b; *SEMA6B*, semaphorin 6b; *DSC1*, desmocollin1; *TNFAIP1*, TNF-alpha-induced protein 1; *MYLK3*, myosin light chain kinase 3; *WAPL*, wings apart-like protein homolog; *ADORA2B*, adenosine A2B receptor; *LOXHD1*, lipoxygenase homology domains 1; *GP6*, glycoprotein VI platelet; *AZIN1*, antizyme inhibitor 1; *NCSTN*, nicastrin.

**Table 5 animals-13-02846-t005:** KEGG pathway analysis results using genes whose somatic mutations were confirmed through WES.

Pathway ID	Name of Pathways	No. of Genes Involved	Name of Genes Involved
hsa04270	vascular smooth muscle contraction	2	*ADORA2B*, *MYLK3*
hsa04020	calcium signaling pathway	2	*ADORA2B*, *MYLK3*
hsa04611	platelet activation	2	*GP6*, *MYLK3*
hsa04512	ECM-receptor interaction	1	*GP6*
hsa04921	oxytocin signaling pathway	1	*MYLK3*
hsa04810	regulation of actin cytoskeleton	1	*MYLK3*
hsa04360	axon guidance	1	*SEMA6B*
hsa04371	apelin signaling pathway	1	*MYLK3*
hsa05034	alcoholism	1	*ADORA2B*
hsa04022	cGMP-PKG signaling pathway	1	*MYLK3*
hsa04971	gastric acid secretion	1	*MYLK3*
hsa05010	Alzheimer’s disease	1	*NCSTN*
hsa04510	focal adhesion	1	*MYLK3*
hsa04330	notch signaling pathway	1	*NCSTN*
hsa04080	neuroactive ligand-receptor interaction	1	*ADORA2B*
hsa04015	Rap1 signaling pathway	1	*ADORA2B*

*ADORA2B*, adenosine A2B receptor; *MYLK3*, myosin light chain kinase 3; *GP6*, glycoprotein VI platelet; *SEMA6B*, semaphorin 6B; *NCSTN*, nicastrin.

## Data Availability

The data presented in this study are available on request from the corresponding author.

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
