# Peer review of "Canine Somatic Mutations from Whole-Exome Sequencing of B-Cell Lymphomas in Six Canine Breeds—A Preliminary Study"

_animals, 2023, doi:10.3390/ani13182846_

Round 1
Reviewer 1 Report
The manuscript “Canine somatic mutations from whole exome sequencing of B-cell lymphomas in six canine breeds” deals with the identification of somatic mutations in dogs affected by BCL.
The manuscript is generally well written and straightforward, the introduction complete and informative and the results clearly reported.
The discussion extensively elucidates the possible role of different genes studied, I think it should also include a few sentences to better comment on one of the main aims of the study, i.e. “…determine the impact of variants from lymph-node aspirate samples compared with whole blood”
I also recommend a careful check of citations and references. E.g.
- ref. 61 is cited in the main text to support the hypothesis of an association between germline mutations and lymphoma. Is ref 61 (Grillet et al., 2009) actually dealing with this topic?
- I can not find ref. 56 in the main text
- Line 336-338 are ref. 57 and 58 correctly reported?
Author Response
We are grateful for Reviewer's dedicated time spent on the review. Regarding the points raised, we would like to provide the following response:
Thank you for the reviewer's advice. As suggested in the beginning of the discussion, we found it necessary to reiterate the identification of variants through a comparison of WB and LN, as you pointed out. Following your guidance, we have revised the opening sentence as follows: "In this study, WES analysis was conducted on both WB and LN aspirate samples from eight dogs with BCL. Based on the results, a comparison between LN aspirate and WB revealed the presence of mutations in LN aspirate samples. This led to the identification of 17 shared somatic variants with moderate or high impact, including GOLIM4, ~."
Despite conducting a final check on the references until the last moment, we made an error in uploading the previous version of the file. As a result, some references were either not mentioned in the main text or appeared to cite incorrect content.
- The reference mentioned as [61] in your comments corresponds to [66], which discusses the relevance of germline mutations in DLBCL.
- The reference cited as [56] at line 320 was submitted in its pre-modified state and is, in fact [61].
- Both [57] and [58] are, respectively, [62] and [63], and both are related to calcium signaling.
The references from the "Discussion" to the end were all incorrectly written, and we have finally corrected them and proceeded with the upload.
Once again, we appreciate the Reviewer's efforts and valuable feedback, and we have made the necessary corrections as per the provided guidance.
Reviewer 2 Report
The manuscript “Canine somatic mutations from whole exome sequencing of B-cell lymphomas in six canine breeds” describes an interesting and innovative study on genetic mutations in canine lymphomas.
The study is well designed and clearly exposed. Material and Methods are fitting with the study and the results obtained are reliable.
The discussion is a little bit long because, of course, includes a long list of genes and their possible role in canine lymphoma, but it is necessary, so I don’t suggest shortening or change it.
In the discussion authors also list the limitations of the study that, anyway, is interesting and opens to further research on lymphoma as on other canine cancers.
However, before listing some minor issue I would like to remark that no ethic statement is included in the manuscript, nor permission number. Owner consensus is not nominated and finally, at the end of the manuscript appears “Informed Consent Statement: Not applicable”.
So that, before considering the manuscript for publication authors are asked to clarify ethical issues.
Clarified this, here below just some comments:
Line 69-71: avoid personalizing the text with “researchers”. Simply say that in this study….DNA samples from…were analysed
Figures: if the manuscript is not examined with pc, the figures are very small. Impossible to read them in a printed version, however this could be not important.
Author Response
We are grateful for Reviewer's dedicated time spent on the review. Regarding the points raised, we would like to provide the following response:
1. The whole blood in this study was the remaining sample collected for the patient's CBC, and the LN sample was the remaining sample collected for the FNA and PARR tests. Due to these reasons, this study did not go through an IRB review. "The Informed Consent Statement" was incorrectly labeled due to our mistake. All the animal owners who visited the veterinary hospital, receive information that all types of samples collected from their animals may be used for future research purposes. After receiving such information, the owner goes through a process of agreement the consent form if they agree to the provided information. So "The Informed Consent Statement" will be changed to "Informed consent was obtained from all owners of dogs involved in the study."
2. The reviewer referred to L69-71, but it might actually pertain to L17-18. The original sentence "The researchers analyzed DNA samples from eight dogs with BCL and performed immunophenotyping using PCR for antigen receptor rearrangement (PARR)." We acknowledge your observation and have revised the sentence "This study analyzed DNA samples from eight dogs with BCL and conducted immunophenotyping using PCR for antigen receptor rearrangement (PARR)."
3. The reviewer thought the figures may be too small and difficult to see. So, we checked through the preprint and didn't encounter significant difficulties viewing them. Thank you for your thoughtful feedback.
Reviewer 3 Report
Dear Editor,
in my opinion, this is a well-written manuscript with all parts of the scientific article. All parts of the manuscript are clear, although the topic is difficult and complicated. In lines 349-357 Authors write about the limitations of this study. These are very important limitations. In my opinion, the small number of dogs and a lot of breeds are the most important. This is why the title of the manuscript must be changed, for example, "Canine somatic mutations from whole exome sequencing of B-cell lymphomas - the preliminary study." I think that the sentence: " Therefore, further studies on germline mutations and CL in more cases of neoplasm in dogs of the same breed will be needed." at the end of the conclusion would be a good ending.
Best regards
Stanislaw Dzimira
Author Response
Thank you for dedicating your time to reviewing our manuscript. We are grateful for your valuable feedback, and we have carefully considered and incorporated the suggested changes as we believe they are essential to be adequately addressed. We have revised the title and the concluding section of the paper according to your advice. Thanks to your guidance, we believe this manuscript has improved.